# Determining the hierarchical order by which intestinal tract, administered diet, and individual relay can shape the gut microbiome of fattening quails

Giovanni Bertoldo[1], Chiara Broccanello[1], Alessandra Tondello[1], Silvia Cappellozza[2], Alessio Saviane[2], Attawit Kovitvadhi[3], Giuseppe Concheri[1], Marco Cullere[4], Piergiorgio Stevanato[1], Antonella Dalle Zotte[4], Andrea Squartini[1]*

1 Department of Agronomy, Food, Natural Resources, Animals and Environment, DAFNAE, University of Padova, Padova, Italy, 2 Council for Agricultural Research and Economics, Research Centre for Agriculture and Environment, Sericulture Laboratory of Padua, Padova, Italy, 3 Department of Physiology, Faculty of Veterinary Medicine, Kasetsart University, Bangkok, Thailand, 4 Department of Animal Medicine, Production and Health, MAPS, University of Padova, Padova, Italy

* squart@unipd.it

**Data Availability Statement:** All relevant data are within the paper and its Supporting information files. The sequences have been deposited in the

## Abstract

A bacterial metabarcoding approach was used to compare the microbiome composition of caecal and faecal samples from fattening Japanese quails (*Coturnix coturnix japonica*) fed three different diet regimes. The tested feedstuffs included (1) a commercial diet for fattening quails, (2) a commercial diet containing 12% full-fat silkworm (*Bombyx mori*) pupae meal, and (3) a commercial diet containing 12% defatted silkworm pupae meal. The aim of the experiment was to verify the relative effect of three variables (diet type, gut tract comparing caecum to rectum, and individual animal) in determining the level of bacterial community dissimilarity to rank the relevance of each of the three factors in affecting and shaping community composition. To infer such ranking, the communities resulting from the high-throughput sequencing from each sample were used to calculate the Bray-Curtis distances in all the pairwise combinations, whereby identical communities would score 0 and totally different ones would yield the maximum distance, equal to 1. The results indicated that the main driver of divergence was the gut tract, as distances between caecal and faecal samples were higher on average, irrespective of diet composition, which scored second in rank, and of whether they had been sampled from the same individual, which was the least effective factor. Simpson's species diversity indexes was not significantly different when comparing tracts or diets, while community evenness was reduced in full-fat silkworm diet-fed animals. The identities of the differentially displayed taxa that were statistically significant as a function of gut tract and diet regimen are discussed in light of their known physiological and functional traits.

## Introduction

Gut microbiomes have been recognized as fundamental in ruling animal physiology across a wide scale of complexity. Evidence includes invertebrates [1], amphibians [2], reptiles [3],

GenBank https://www.ncbi.nlm.nih.gov/ SRA (Sequence Reads Archive repository under the code PRJNA975099).

**Funding:** This research was supported by the University of Padova (Italy) funds (2019 -DOR1923890/19) and by funds of the project "Serinnovation - Dimostrazione d'innovazione, qualità,16 tracciabilità in gelsibachicoltura per lo sviluppo di fonti integrative per le aziende agricole - Misura 16.1 -2 - Programma di sviluppo rurale per il Veneto 2014-2020 - DGR 2175 del 23/12/2016. The funders had no role in study design, data collection and analysis, decision to publish, or preparation of the manuscript.

**Competing interests:** The authors have declared that no competing interests exist.

birds [4], mammals [5], and a vast body of studies on the consequences of microbiome diversity and stability physiology and health in humans [6]. When trying to assess the effects of external factors over those exerted by individual or species genetics, the debate is still open regarding whether the former or the latter prevails in shaping the gut microbiota composition, either in human subjects or across several vertebrate clades [7–9].

Factors that have been demonstrated effective in affecting microbiome composition include environmental ones, as biogeographical context and population social structure. Others are related to the specific individual, including sex and reproductive condition [10]. But the major drivers consistently pointed out by independent studies are the evolutionary history of the host as fixed constraint and its diet as the variable factor [11, 12].

Beyond genetic legacies, diet is recognized as one of the most profoundly affecting factors in determining microbiota composition, which can be rapidly changed upon diet shifts [13, 14]. However, the fact that each individual would possess, and possibly retain, a certain degree of uniqueness in its inherited microbiome and a genetically constrained constancy in its composition, in spite of external factors, has also emerged from different evidence [15, 16].

At a further complexity level lies the fact that, within a certain individual, in addition to the factors of taxonomy, genetics and diet, microbiome composition also reflects variability related to which specific tract of the gut is analyzed [10, 17]. Both reports supported the evidence that bird caecal and faecal communities are therefore rather distinct, which is one of the premises for the comparative approach that we adopted in the present work. Other authors quantified the level of difference between caecal and faecal samples in broiler chickens. Data indicated that cloacal communities were more tightly related to caecal communities when the paired analyses involved the same individual compared with random pairs of caecal vs. faecal samples of different chickens from the same flock [18].

In the present report, capitalizing from our prior investigations on different diet digestibility in quails [19] and exploiting the same setup, we extended the microbiological analysis by sequencing in the caeca and comparing the bacterial communities found in the two compartments (caeca vs. faeces). The following question was addressed: in what hierarchical order do variables act in shaping the composition of microbial communities? The variables considered were: (1) intestinal tract: caecum or rectum, (2) diet: control diet, 12% full-fat silkworm meal, or 12% defatted silkworm meal, and (3) individual quail. To address this query, we determined the compositional dissimilarity between each community in pairwise comparisons, expressed it as the Bray-Curtis ecological distance index, and then compared the means of each group pertaining to the variable definition. Further analyses of the data involved multivariate statistics ordination and computation of ecological indexes.

## Materials and methods

### Ethics statement

All experimental protocols were approved by the Ethical Committee for Animal Experimentation of the University of Padova (Ethical approval number 147882). All methods were carried out in accordance with the relevant guidelines and regulations: the animals were handled in compliance with the principles stated by the EC Directive 86/609/EEC regarding the protection of animals used for experimental and other scientific purpose. All the methods are reported, in accordance with the ARRIVE guidelines for in vivo research involving animals (https://arriveguidelines.org/).

## Faecal and caecal microbiome analysis

The excreta and the corresponding caecal content of 15 quails (n = 5/dietary treatment) were the source for total DNA isolation. The three dietary treatments were (1) a commercial diet for fattening quails (control diet), (2) a commercial diet containing 12% full-fat silkworm (*Bombyx mori*) pupae meal (full-fat SW), and (3) a commercial diet containing 12% defatted silkworm pupae meal (defatted SW). The detailed farming protocol can be found in [19]. Briefly, 16 day-old, separately-caged quails were assigned to the different dietary regimes and the DNA extraction from the individual excreta and caeca took place after 15 days since the beginning of the feeding trial. DNA extraction was carried out upon homogenizing 100 mg of freshly collected material using a Tissue Lyser (Qiagen, Hilden, Germany) for 5 min at 30 Hertz in a 2 ml Eppendorf tube with 300 μl of RTL buffer (Qiagen, Hilden, Germany). Samples were centrifuged for 5 min at 6,000 g, at 5˚C to collect the supernatant. DNA purification was performed with a Biosprint 96 suite using the MagAttract HMW DNA Kit (both from Qiagen, Hilden, Germany) as recommended by the manufacturer. DNA was quantified with a Qubit 3.0 Fluorometer using the Qubit™ DNA HS Assay Kit Fluorometer (Thermo Fisher Scientific, Waltham, MA, USA). Purified DNA was kept at -20˚C until sequencing.

A multi-amplicon sequencing strategy) targeting V2, V3, V4, V6-7, V8 and V9 hypervariable regions was adoptedusing the Ion 16STM Metagenomics multi-amplicon Kit (Thermo Fisher Scientific, Waltham, MA). Library preparation was carried out using the standard protocol recommended by the manufacturer. Sequencing was performed with an Ion™ Torrent S5 System with a 520 chip using 850 flows. The uBAM files sourced from the Ion GeneStudio platform were converted into FASTQ format using the samtools bamtofastq (v1.10). A 20-nucleotide trimming on both ends of the raw reads was performed to eliminate the sequencing primers using cutadapt (v3.5). A "Quantitative Insights Into Microbial Ecology 2" (QIIME2) (v2020.08) pipeline was subsequently used to analyse the trimmed raw reads. Within this process, imported reads were denoised and dereplicated using the "qiime dada2" plugin followed by OTU clustering with a 97% sequence similarity cutoffs using the Qiime vsearch plugin. The representative sequences from OTUs were then classified using SILVA SSU (version 138.1) as the reference database.

Molecular data regarding bacterial species compositional differences across the different treatments were analyzed using the Calypso online software tool [20]. The relative abundances of taxa were normalized by applying total sum of squares scaling (TSS) normalization followed by square root transformation or centered log ratio (CLR) transformation, depending on the analysis. Significant differences among treatments ($P < 0.05$) were assessed by ANOVA or by the nonparametric Kruskal-Wallis test.

The sequences have been deposited in the GenBank https://www.ncbi.nlm.nih.gov/ SRA (Sequence Reads Archive repository under the code PRJNA975099.

## Community distance analysis

The ranking of the three variables (intestinal tract, diet type, individual animal) as drivers of bacterial community structure was based on the analysis of the compositional difference of these among each category. In ecology, the Bray–Curtis distance [21] is a statistical index, as well as a normalization method, used to quantify the compositional dissimilarity between two different communities based on counts of shared or unique species and taking into account presence and abundance. The advantage of this index over other measures is that it suits comparisons in which the matrix may also feature a conspicuous number of values equal to zero, i.e., a taxon being absent in one or more communities. This is often the case in microbial checklists and when comparing habitats that can be differentially colonized or far apart from

each other. First we computed the 406 pairwise crossed comparisons (= total number of replicates for the analysis) yielding the Bray-Curtis distances calculated using the PAST Software [22]. The means of the distances of the groups (same diet vs. different diets; same tract vs. different tracts, same individual vs. different individuals), and their various combinations were compared to establish the difference imparted by each of the three factors. To achieve the ranking of the three variables (intestinal tract, diet type, individual animal), from the resulting beta diversity matrix of Bray-Curtis distance values, the orderly method that we adopted was to cut out subsets of the resulting data to separately consider each aspect. For this purpose, upon cutting sections of the whole matrix in Microsoft Excel, we calculated the mean distances, their standard deviations (SD) and the ensuing coefficient of variation CV (SD/mean), working on 34 submatrixes that correspond to each of the possible diet, or tract or animal combination groups. The resulting dataset was then ordered by the values of the means column in decreasing order to obtain a distribution gradient and interpret the relative effect of the three variables (tract, diet, individual animal) contributing to the ranking.

## Results

The 16S bacterial amplicon community sequencing, after denoising, quality filtering chimera checking, OTU picking and annotation, yielded a total of 1,144,642 filtered high-quality sequences, with an average of 3,9470.41 sequences per sample. A total of 374,403 sequences were from the faecal samples, and 770,239 sequences were from the caecal samples.

A total of 124 taxa were identified to species-level rank accuracy. Twenty-nine of these were only encountered in caeca, 35 only in faeces and 60 were shared by both tracts. The unique ones were in all cases minority occurrences that did not exceed 0.6% of the reads and in most instances were of far lower numerical relevance in both tracts. A cluster analysis of the communities based on the Bray-Curtis distances generated the dendrogram shown in Fig 1. Three

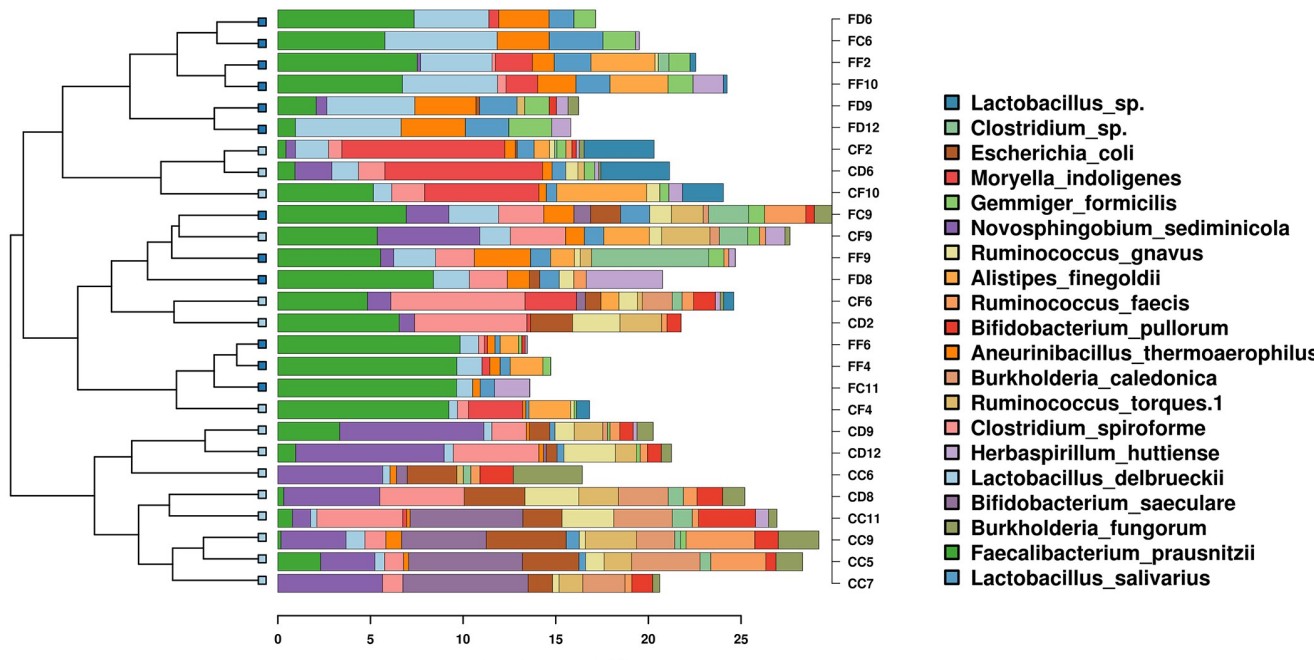

**Fig 1. Cluster analysis dendrogram based on the Bray-Curtis community distance data for the top 20 most abundant taxa.** (Square root/total sum of squares data transformation). The bacterial community partitioning is shown (left), along with the identities of the featured taxa (right). Sample codes: First letter: gut Tract (C = Caeca, F =: Faeces); second letter: Diet (C = control, D = defatted SW meal, F = full-fat SW meal); Digits: individual quail number.

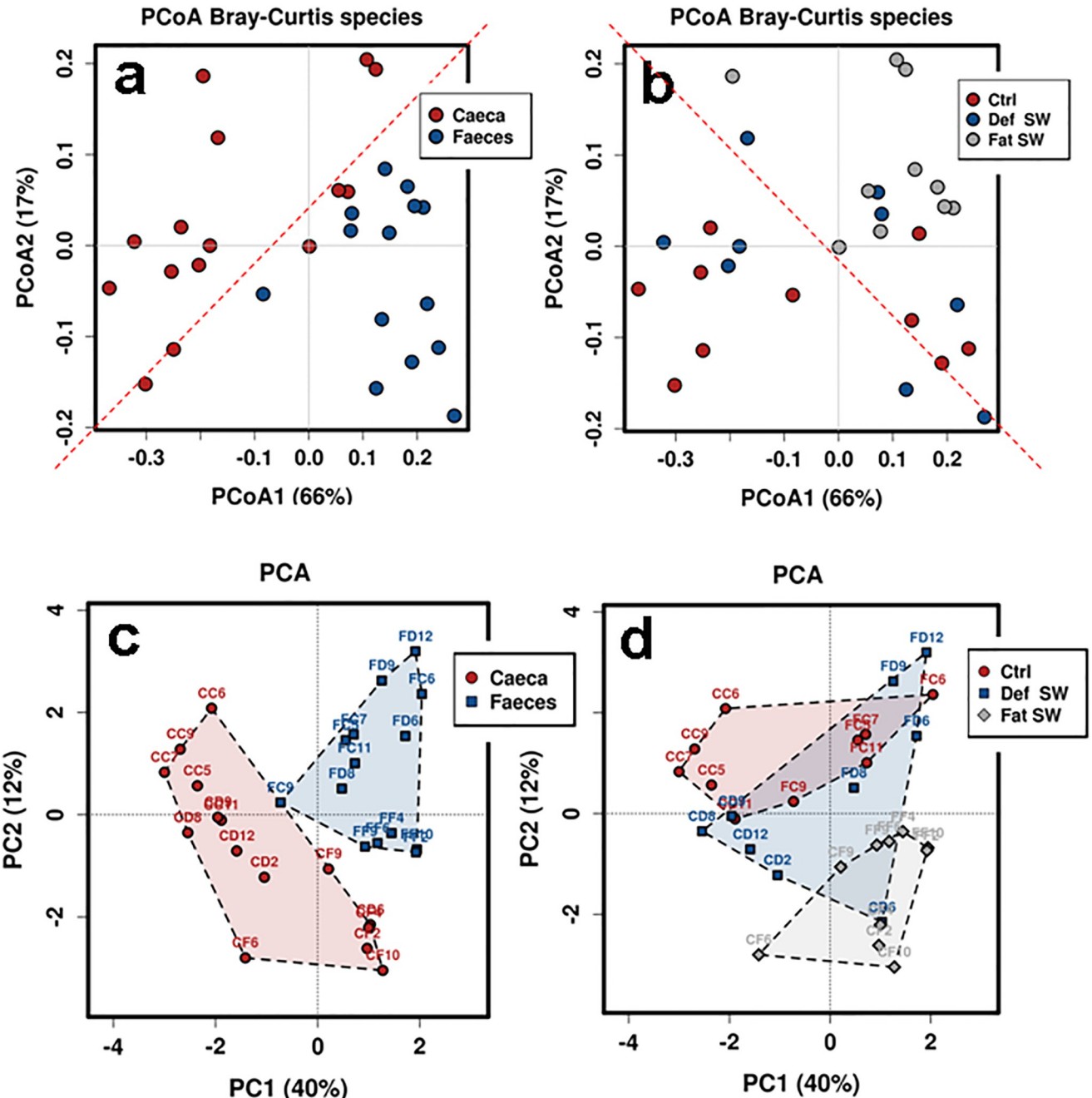

**Fig 2. Multivariate ordinations of the bacterial community presence and abundance data.** Principal Coordinate Analysis (a, b) and Principal Component Analysis (c, d) are compared. Sample labeling is shown according to gut Tract (a, c) or to Diet (b, d).

main clusters were individuated, one of which (the bottom cluster in Fig 1) included only cae-
cal communities, and the other two included a prevailing presence of faecal communities.

Multivariate ordination (Fig 2) using Principal Coordinate Analysis (PCoA) or Principal
Component Analysis (PCA) showed a clear partitioning between the two gut tracts and a less
pronounced separation among the three diets, confirming the following hierarchy:
Tract > Diet. In PCoA the diagonal axis in panel a is drawn to demonstrate the split between

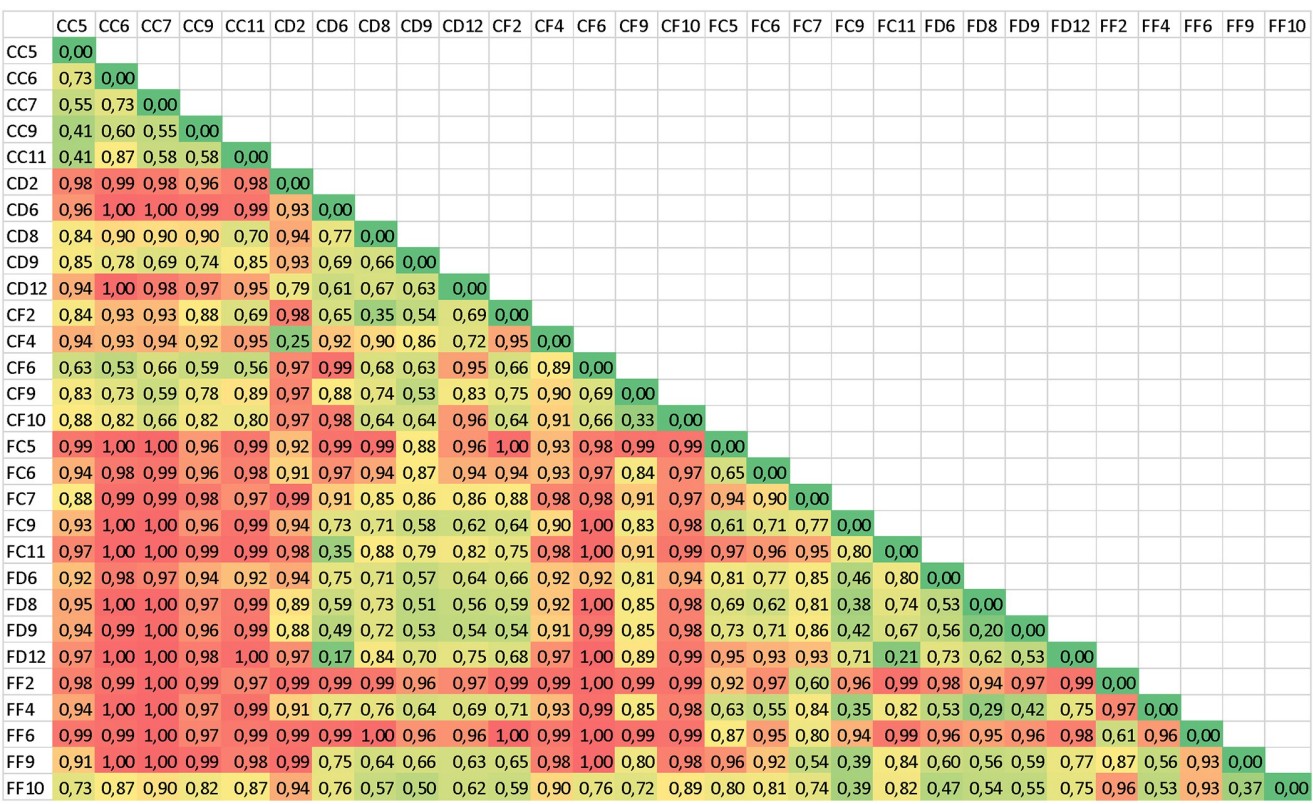

**Fig 3. Bray-Curtis distances between each of the bacterial communities.** All pairwise comparisons between the different communities based on the presence and abundance of the taxa were calculated. Conditional formatting from red (highest distances) to green (lowest distances) allows us to spot the gradient of similarities. Sample codes: First letter: Tract (C = Caeca, F = Faeces); second letter: Diet (C = control, D = defatted SW meal, F = full-fat SW meal); Digits: individual quail number.

caeca and faeces (Fig 2a). The specular axis in panel b instead separates the upper right sector, in which all communities fed with the full-fat silkworm diet ended up, from the bottom left half where only cases fed with the other two types were featured (Fig 2b). Principal component analysis (Fig 2c and 2d) consistently showed that the control and defatted SW diets resulted in more overlap than the full-fat SW diet.

To numerically compare the individual differences among samples, we used the computed Bray-Curtis differences matrix. To apply such metrics, all pairwise comparisons between the bacterial communities of each sample were run. The minimum (identical communities) yield zero (0), as in the self-comparison diagonal line, while the theoretical limit of 1 would represent the maximum of the difference that could occur between totally unrelated communities.

Results are shown in Fig 3. Observing the conditional formatting, a consistently higher distance (red shades) can be spotted along the first columns between the CC samples (caeca, control diet) and most of the faecal samples (codes starting with F).

As described in Materials and Methods paragraph 2.2, upon cutting sections of the whole matrix shown in Fig 3, we calculated the mean distances, their standard deviations and the ensuing coefficient of variation of the 34 submatrixes that correspond to the groups listed in Fig 4, in which the row labeled 'ALL' is the one expressing the average community distance resulting from the whole project (the mean from all comparisons considered, as shown in Fig 3). Each of the other rows instead shows the results obtained by selecting values that shared

| | Mean | SD | CV |
|---|---|---|---|
| Same tract (Faeces) Same diet (SW Defatted) | 0,53 | 0,18 | 0,34 |
| Same tract (Caeca) Same diet (Control) | 0,60 | 0,14 | 0,24 |
| Different tract Same diet (SW Defatted) | 0,67 | 0,19 | 0,28 |
| Same diet all (any tract) (SW Defatted) | 0,67 | 0,19 | 0,28 |
| Different tract SAME QUAIL (SW Defatted) | 0,69 | 0,11 | 0,16 |
| Same tract (Faeces) Different diet (FC-FD) | 0,70 | 0,20 | 0,28 |
| Same tract (Faeces) Different diet (FD-FF) | 0,73 | 0,23 | 0,31 |
| Same tract faeces (any diet) | 0,74 | 0,21 | 0,28 |
| Same tract (Caeca) Same diet (SW Fat) | 0,74 | 0,19 | 0,25 |
| Same tract (Caeca) Same diet (SW Defatted) | 0,76 | 0,13 | 0,17 |
| Same tract all (any diet) | 0,76 | 0,19 | 0,25 |
| Same tract (Faeces) Same diet (SW Fat) | 0,77 | 0,23 | 0,30 |
| Same tract (Caeaca) Different diet (CD-CF) | 0,77 | 0,21 | 0,27 |
| Same tract (Faeces) Different diet (FC-FF) | 0,78 | 0,20 | 0,26 |
| Same tract caeca (any diet) | 0,79 | 0,17 | 0,22 |
| Same tract (Caeca) Different diet (CC-CF) | 0,79 | 0,14 | 0,17 |
| Same tract (Faeces) Same diet (Control) | 0,83 | 0,14 | 0,17 |
| Different tract Different diet (CD-FF) | 0,83 | 0,17 | 0,20 |
| --- ALL ------------------------------------ | 0,83 | 0,18 | 0,22 |
| Same diet (all) (any tract) (SW Fat) | 0,84 | 0,18 | 0,21 |
| Different tract Different diet (CD-FC) | 0,85 | 0,15 | 0,18 |
| Same diet (all) (any tract) (Control) | 0,86 | 0,18 | 0,21 |
| Different tract Same diet (all) | 0,86 | 0,18 | 0,21 |
| Different tract Different diet (CF-FD) | 0,87 | 0,14 | 0,16 |
| Different tract SAME QUAIL (Caeca-Faeces (any diet) | 0,88 | 0,15 | 0,17 |
| Different tract (all) (any diet) | 0,89 | 0,15 | 0,17 |
| Different tract Different diet (all) | 0,90 | 0,13 | 0,14 |
| Different tract Same diet (SW Fat) | 0,91 | 0,13 | 0,14 |
| Same tract (Caeca) Different diet (CC-CD) | 0,91 | 0,10 | 0,11 |
| Different tract SAME QUAIL (Caeca-Faeces (SW Fat) | 0,92 | 0,08 | 0,09 |
| Different tract Different diet (CF-FC) | 0,93 | 0,09 | 0,09 |
| Different tract Different diet (CC-FF) | 0,95 | 0,07 | 0,07 |
| Different tract Different diet (CC-FD) | 0,97 | 0,03 | 0,03 |
| Different tract Same diet (Control) | 0,98 | 0,03 | 0,03 |
| Different tract SAME QUAIL (Caeca-Faeces (Control) | 0,98 | 0,01 | 0,01 |

**Fig 4. Mean of all the pairwise community comparisons (Bray-Curtis ecological distances) grouped by variable.** The table is ranked vertically by increasing order of Bray Curtis distance mean.

either the same gut tract (green entries) or were taken from a different tract (red entries). The other combinations considered the three diets and each set of conditions. Ordering the values of the means column in decreasing order yielded the distribution shown in Fig 4 that allows to interpret how the three variables contributed to the ranking. The result shows that the "Same tract' sets prevailingly occur in the part of the table that is above the experiment's mean ('ALL' dotted line—-). This upper portion of the table corresponds to the highest community similarity (lowest Bray-Curtis distance means), with the exception of the silkworm defatted diet cases, which, irrespective of the tract, accumulate in the upper section. It is also visible that the highest CV are observed in relation to the highest similarities, while the lowest (blue shades) correspond to the most dissimilar community comparisons. This indicates that similar communities at the same time tend to contain more variability in their data.

The post-hoc calculation of the statistical power that applied to our analyses, given the total number of replicates (406 pairwise comparisons of beta-diversity among samples), the means and SD recorded and the number of replicates belonging to each different subsets, resulted, aiming at a statistical $P$-value threshold of $< 0.05$ to be spanning between 98.1% for the subsets featuring the highest difference between cases (same tract same diet vs. all) to 100% for the subsets featuring the smallest difference between cases (same tract-any diet vs. different tract-any diet).

To dissect the hierarchy into finer detail and to inspect the variability within each subset, we presented the data from the same table of Fig 4 rearranged in five separate tables, which are combined in S1 Fig (S1 File). In that case, the panels show the communities by tract, with the two sectors of the digestive system analyzed. Those across heterologous tracts only, those by diet type and those within individual animal. These consist of the means of each pair of comparisons that concern the caecum and faeces of the same quail. In each panel, data are still arranged in decreasing order of the community distance means.

Sideways, close to each of the same panels, the order by which the three different diets resulted ranked is shown. In the faeces, the diet that mostly keeps communities similar among the five replicates is defatted SW, followed by full-fat SW, and last by the control. In caeca, the situation is completely and symmetrically the opposite (control > full-fat SW > defatted SW).

Additionally, when comparisons are heterologous, as shown in the third panel of S1 Fig (across tracts) in S1 File, the diet that maintains more similarities of the passing microbiomes is the defatted SW, over full-fat SW and then control. The same ranking was also confirmed by the three diet comparisons (fourth panel from the top), irrespective of the tract: defatted SW > full-fat SW > control. The same order is also yielded by the comparisons between the two tracts within the same individual quail (fifth panel): defatted SW > full-fat SW > control.

These data consistently concur to suggest that the hierarchical order by which the three variables act results in the following:

Intestinal tract > Diet composition > Individual animal factor

In addition to establishing this order, the other goals of this research were a) to unravel the taxonomic identities of the dominant species of fattening quails and investigate their possible roles on the basis of known physiological traits from microbial ecology literature, b) to trace the specificity within the two tracts and c) to point out possible diet-related shifts. The results are in the form of annotated identities of the sequences stemming from the bioinformatics analysis.

The percentages hereby reported are the relative abundance for each taxon within the whole sequencing output table. The dominant taxa in the caecal samples were *Lactobacillus delbrueckii* (26.5% of the reads) and featured almost exclusively in the full-fat SW diet. The second most abundant was *Faecalibacterium prausnitzi* (17.4%), widespread across all three diet

regimes, and the third was *Lactobacillus salivarius* (13.4%). This latter taxon resulted in the overwhelmingly most abundant sequence in the faecal samples, where it accounted for 62.4% of the sequences in terms of relative abundance and occurred throughout all diets. The second in the faecal scores was *Burkholderia fungorum* (11.37%), and the third was *Herbaspirillum huttiense* (3.75%), both of which were abundantly featured in all three diets.

To define which taxa were differentially present in statistically significant proportions between the caeca and the faeces, a nonparametric Wilcoxon rank test analysis was run on the presence/abundance matrix yielding the results shown in Fig 5. The latter lists the 14 taxa that

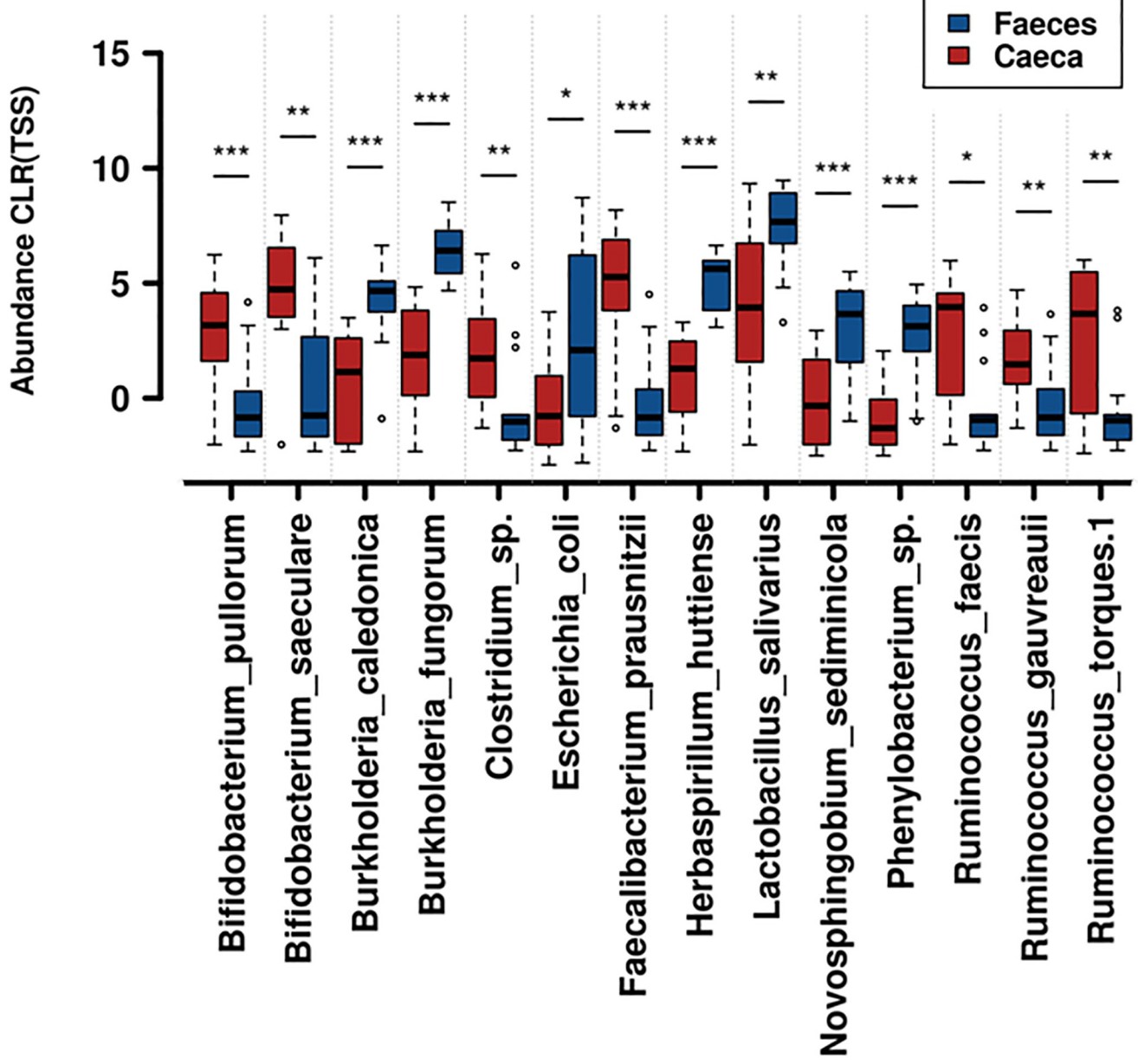

**Fig 5. Differentially featured taxa by intestinal tract.** Rank test plot based on pairwise comparisons by Wilcoxon rank test of the taxa that were significantly different across the comparison between caecum and faeces. Compared data were transformed by Centered Log Ratio/Total Sum of Squares (CLR TSS). Significant differences are marked by *: P < 0.05; **: P < 0.01; ***: P < 0.001. Red bars: caecal communities; blue bars: faecal communities.

stood out, six of which were significantly more abundant in faeces, and the remaining eight were significantly more abundant in caeca. The relative position of the blue bar compared to the red bar in each paired comparison of Fig 5 allows to identify which taxa belong to which of these two categories.

Regarding the diet type, the corresponding analysis results are plotted in Fig 6. Nine taxa passed the cutoff of significance levels for differences between at least one diet in comparison to either one or both of the other two diets. The full-fat SW diet was significantly enriched with four bacterial taxa with respect to the other diets (*Aneurinibacillus thermoaerophilus*, *Bacillus thermoamylovorans*, *Lactobacillus delbrueckii*, and *Lactobacillus* sp.).

All cases featured in the rank tests of Figs 5 and 6 also belong to the high end of the abundance scores and represent in most cases dominant species of those communities in one or more of the tracts and diet compared. Their potential roles in animal physiology are reviewed in the discussion section. Further details on these and other less abundant taxa, but displaying statistically significant differences, are shown in S2-S4 Figs in S1 File.

Finally, the ecological indexes of Simpson's diversity and community evenness were not significantly different between the two gut tracts (S5 Fig in S1 File). The SW diets did not significantly affect diversity; however, a decreasing trend was appreciable from the data for the full-fat SW diet, and this diet significantly reduced community evenness, indicating its possible excessive burden on microbiome stability in comparison to its defatted version.

## Discussion

To frame the focus of this comparison it is worth recalling that in birds, the role of caeca involves not only the reabsorption of water and salts but also fermentation of undigested carbohydrates and uric acid, thereby re-cycling nitrogen and generating ammonia and volatile fatty acids. Conversely, the rectum, from which the faeces are ejected, is a terminal passage with no specific physio-metabolic functions.

We addressed the order by which the analyzed variables act in determining the assemblage of bacteria found in the different gut compartments. The first remarkable evidence was that, as documented in Fig 4, communities in the same tract have much more affinity than communities shaped by the same diet. The descending ranking based on community similarity is ruled vertically by "Same tract" over "Different tract". Moreover, the coefficient of variation was inversely proportional to the community similarities, indicating that samples from the same tract also have less variability of composition.

Regarding the three diets tested, the full-fat SW diet appears to confer some effects of globally unifying communities despite their being in different tracts. This is suggested by the fact that the rare instances of "Different tract" featured in the upper part of the table are due to the defatted SW diet.

Thus, considering that those two gut tracts bear the major differences in bacterial community composition, one can inspect the identities of the key differentially featured taxa and the implications that can be drawn from available knowledge on their physiology and ecology. In exploring this aspect, it can also be considered that the caeca are dead-ended pockets of food processing, which qualify more typically as reservoirs to host more stably resident biota involved in nutrition-related processing, while on the contrary, faeces mirror straightly the rectum composition and embody an expelled community [17]. The latter could contain both remnants of the bacteria ingested along with the food itself or their residual DNA, as well as a load of other members of the gut microbiome undergoing physiological turnover [10]. This topological distinction of the two investigated tracts implies that caeca could be envisaged as the site hosting more permanent 'workers' of the quail inner microbiome. Faeces instead, are,

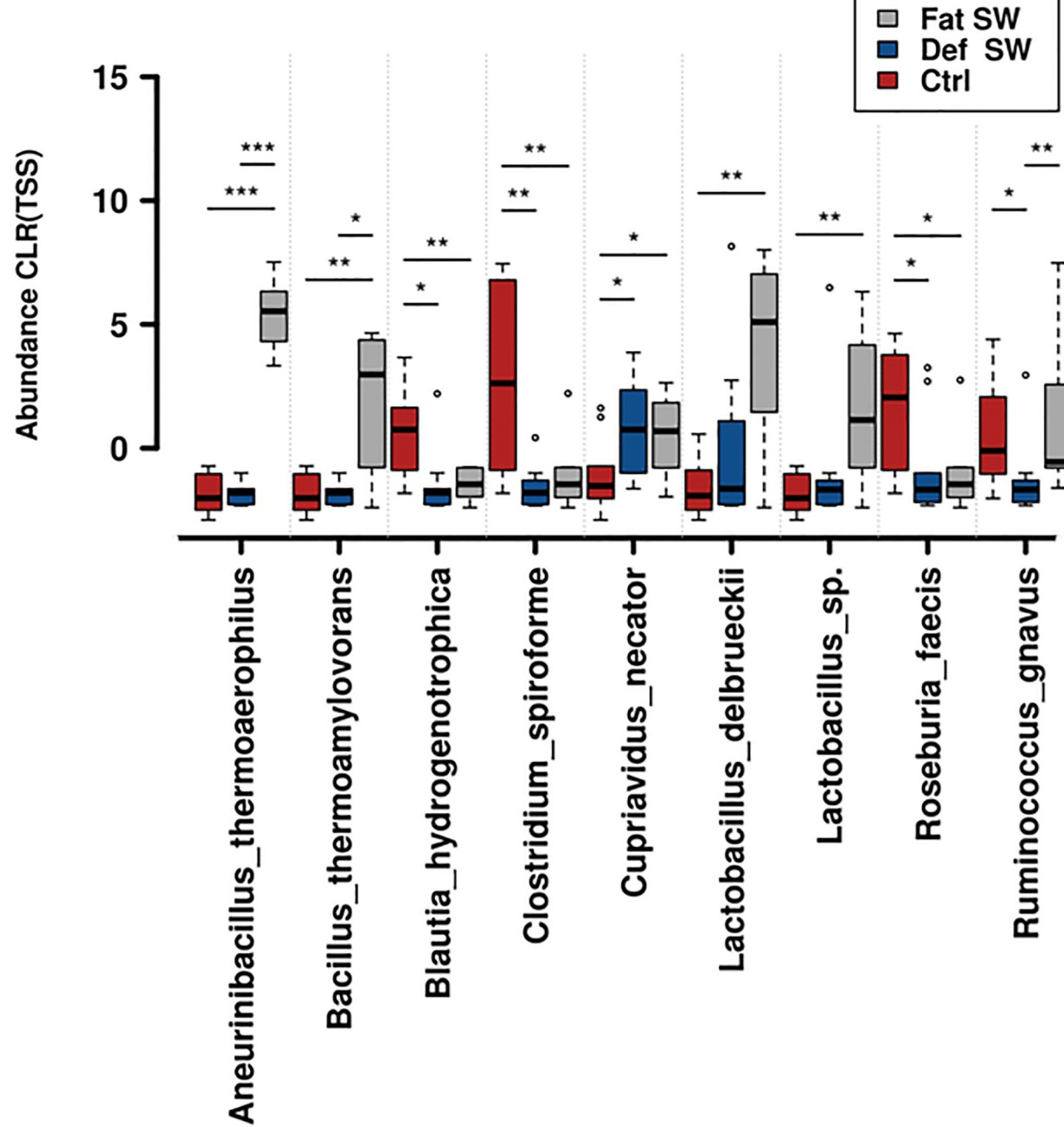

**Fig 6. Differentially featured taxa by diet regime.** Rank test plot based on pairwise comparisons by Wilcoxon rank test of the taxa that were significantly different across the different diet treatments. Compared data were transformed by Centered Log Ratio/Total Sum of Squares (CLR TSS). Significant differences are marked by *: $P < 0.05$; **: $P < 0.01$; ***: $P < 0.001$. The order of treatments for each species is, from left to right, control (Ctrl), defatted SW meal (Def SW), and full-fat SW meal (Fat SW).

by their nature, testifying a transient status of their contained 'migrant' and/or 'dismissed' microbial members.

When reporting and discussing taxa at species level as results of metabarcoding, caution is to be exerted due to the incomplete status of existing databases that led to the annotation. In

this respect our confidence is supported by the sequencing strategy used, as the library preparation hereby adopted targets seven out of nine of the existing hypervariable regions within the 16S rRNA gene, while the standard in the literature is currently still limited to only one of them (V3-V4) [23].

Inspecting the taxa that stood out as significantly different under a statistical grid, regarding the gut tract (Fig 5), starting from the caeca, two prominent cases belong to the *Bifidobacterium* taxon. This genus of oligosaccharide-fermenting actinobacteria is known for its active role in host metabolism, for which it also represents a probiotic microorganism associated with physiological equilibria [24]. The two species observed hereby, *B. pullorum* and *B. saeculare*, were described in early studies from isolates of Italian origin in farm animals such as chicken [25] and rabbits [26]. Another taxon that is abundantly featured in caeca and less abundant in faeces is *Clostridium* sp. Clostridia encompass vast diversity, while collectively, their class is linked to positive effects on gut homeostasis as commensals, some species can represent exceptions [27]. Moving further, we encountered *Faecalibacterium prausnitzii* as significantly more distinctive of caeca than faeces. Additionally, this member of the Ruminococcaceae family is renowned for its positive indications in gut health and probiotic role, being reported as potentially preventing Chron's disease and inflammatory bowel, through production of butyrate, and by influencing gut physiology via secretion of mucous O-glycans [28]. Three other members of the same family were differentially displayed and prevailed in caeca: *Ruminococcus faecis*, able to ferment glucose, lactic and acetic acids [29]; *Ruminococcus gauvreauii*, which is reported as fermenting aesculin and glucose [30]; and *Ruminococcus torques*, which is a species that has been pointed out as a putative marker for potential gut disorders induced by stress conditions [31].

Instead, among microorganisms that resulted significantly more abundant in faeces, two members of the *Burkholderia* genus were found (*B. caledonica* and *B. fungorum*); this group, unlike the various cases just described as prevailing in the caeca, represents obligate aerobic species, which are often isolated from soils and different plant-associated environments [32]: Such instance qualifies it as a transient bacterium rather than e member of the true animal gut-dwelling taxa. Another case with a higher mean value in faeces is *Escherichia coli*, primarily associated with a wide array of animal guts but, at the same time, one of the two major indicators of faecal contamination of water, being expelled daily at rates between 10 and 100 million colony forming units in humans [33]. *Herbaspirillum huttiense* is another microorganism that has no apparent advantage of being retained in the gut. In fact, it is reported as a potential opportunistic pathogen not physiologically involved in host metabolism [34]. *Novosphingobium sediminicola* is also a species with more environmental than gut-associated attitudes that was isolated from freshwater [35]. Finally, in faeces, there is a prevalence of *Phenylobacterium* sp., a strictly aerobic genus that is known for its growth on compounds of industrial origin, which thus has little relation with stable animal microbiota [36]; its transient presence in these farmed animals could be hypothesized as a possible clue of waste residues in poultry feedstuff.

Moving down to the second hierarchy level, the Diet variable, we evaluated its effects in determining upshifts or downshifts in defined taxa for one or two of the three diets with respect to the remaining one(s). The data presented in Fig 6 indicate nine significantly supported taxa. Three of these were promoted by the full-fat SW meal diet in comparison to both the control and the defatted SW meal diets. The three bacterial annotations were *Aneurinibacillus thermoaerophilus*, *Lactobacillus delbrueckii* and *Bacillus thermoamylovorans*. There is a clue that can explain their prevalence; we had previously reported that SW meal contains a compound (1-deoxynojirimycin, 1-DNJ) originating from mulberry leaves, which in turn are used to feed SW larvae. Such molecule inhibits glycosidase activity and results in starch accumulation in the quail digestive tract [19]. In the same work, we described how this causes an

accumulation of undigested starch. The three microorganisms that appear selectively enhanced by the full-fat SW diet to quails are all known for their proficiency to grow on this carbohydrate source, as typical for the *Lactobacillus* group [37], or even recalled by the name (*B. thermoamylovorans* = thermophilic starch-eating). They are also known to grow on glucose resulting from starch partial catabolism, as in the case of *A. thermoacrophilus*, which is usually also isolated on sugarbeet [38]. The fact that their prevalence attains significance on the sole full fat diet can be hypothesized to be due to the fact that the defatted one, lacking the richest sources of energy, would not sustain an equally efficient microbial growth.

In contrast, three species were downshifted in the two SW diets with respect to the control. *Blautia hydrogenotrophica* belongs to a promising probiotic genus encountered in healthy microbiomes of different organisms [39]. *Clostridium spiroforme* is present in caecal flora, although it includes some potentially pathogenic strains [40]; a further case is *Roseburia faecis*. The genus is part of the regular intestinal commensals that produce short-chain fatty acids and positively affect immunity and anti-inflammatory status. Its proportional alteration has been found to be associated with dysbiosis-related disorders and has thus been proposed as a potential marker of human health [41]. Its decrease in both SW-containing diets could suggest a possible alteration of the equilibrium due to components associated with SW meal inclusion.

One taxon was significantly decreased with the defatted SW diet, while it remained stable in both the control and full-fat SW diets and that is *Ruminococcus gnavus*. It is a well-known indicator of negative conditions in human models where it has been associated with Crohn's disease and produces an inflammatory polysaccharide [42]. In this sense, the defatted SW diet would qualify as preferable to the full-fat one in terms of microbial community potential alterations from its regular state.

Only one species was instead significantly enhanced by both the full-fat and defatted SW diets in comparison to the control. This was the case for *Cupriavidus necator*, which is an interesting occurrence, being a hydrogen-oxidizer bacterium able to use both organic compounds and hydrogen. A study comparing healthy and colon cancer-affected human patients reported that this taxon prevailed in control subjects [43].

## Conclusion

Three variables were studied in relation to the fattening quail gut microbiome. These included anatomical location, administered diet, and single individual identity. The hierarchical order by which they act in determining bacterial community structure was assessed. The ranking indicated the gut compartment as the ruling determinant, with the diet type as second force, and, in suborder, the individual animal relay.

## Supporting information

**S1 File.**
(DOCX)

## Acknowledgments

The authors thank Rino Cailotto ("La Colombara" Società Agricola, Castelnovo di Isola Vicentina, VI, Italy), Quaja Veneta® Società Cooperativa Agricola (Malo, VI, Italy), and the feed company FANIN Srl (San Tomio di Malo, VI, Italy) for their crucial support for a successful research project.

## Author Contributions

**Conceptualization:** Silvia Cappellozza, Marco Cullere, Antonella Dalle Zotte.

**Data curation:** Giovanni Bertoldo, Chiara Broccanello, Alessandra Tondello, Andrea Squartini.

**Formal analysis:** Giovanni Bertoldo, Chiara Broccanello, Alessandra Tondello, Andrea Squartini.

**Funding acquisition:** Attawit Kovitvadhi, Giuseppe Concheri, Piergiorgio Stevanato, Antonella Dalle Zotte.

**Investigation:** Giovanni Bertoldo, Chiara Broccanello, Alessandra Tondello, Andrea Squartini.

**Methodology:** Andrea Squartini.

**Project administration:** Silvia Cappellozza.

**Resources:** Giuseppe Concheri, Piergiorgio Stevanato.

**Supervision:** Silvia Cappellozza, Antonella Dalle Zotte, Andrea Squartini.

**Validation:** Silvia Cappellozza, Antonella Dalle Zotte, Andrea Squartini.

**Writing – original draft:** Andrea Squartini.

**Writing – review & editing:** Giovanni Bertoldo, Chiara Broccanello, Alessandra Tondello, Silvia Cappellozza, Alessio Saviane, Attawit Kovitvadhi, Giuseppe Concheri, Marco Cullere, Piergiorgio Stevanato, Antonella Dalle Zotte, Andrea Squartini.

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
