## [Decision Letter · Decision Letter 0]

13 Dec 2023

PONE-D-23-33570Determining the hierarchical order by which the variables of intestinal tract, administered diet, and individual relay can shape the gut microbiome of fattening quailsPLOS ONE

Dear Dr. Squartini,

Thank you for submitting your manuscript to PLOS ONE. After careful consideration, we feel that it has merit but does not fully meet PLOS ONE’s publication criteria as it currently stands. Therefore, we invite you to submit a revised version of the manuscript that addresses the points raised during the review process.

We look forward to receiving your revised manuscript.

Kind regards,

Ewa Tomaszewska, DVM Ph.D

Academic Editor

PLOS ONE

Journal Requirements:

"University of Padova (Italy) funds (2019 - DOR1923890/19) and funds of the project "Serinnovation - Dimostrazione d'innovazione, qualità, tracciabilità in gelsibachicoltura per lo sviluppo di fonti integrative per le aziende agricole - Misura 16.1 -2 - Programma di sviluppo rurale per il Veneto 2014-2020 - DGR 2175 del 23/12/2016” 

"This research was supported by the University of Padova (Italy) funds (2019 -DOR1923890/19) and by funds of the project "Serinnovation - Dimostrazione d'innovazione, qualità,16 tracciabilità in gelsibachicoltura per lo sviluppo di fonti integrative per le aziende agricole - Misura 16.1 -2 - Programma di sviluppo rurale per il Veneto 2014-2020 - DGR 2175 del 23/12/2016"

 "University of Padova (Italy) funds (2019 - DOR1923890/19) and funds of the project "Serinnovation - Dimostrazione d'innovazione, qualità, tracciabilità in gelsibachicoltura per lo sviluppo di fonti integrative per le aziende agricole - Misura 16.1 -2 - Programma di sviluppo rurale per il Veneto 2014-2020 - DGR 2175 del 23/12/2016” 

Reviewers' comments:

Reviewer's Responses to Questions

**Comments to the Author**

1. Is the manuscript technically sound, and do the data support the conclusions?

Reviewer #1: Yes

2. Has the statistical analysis been performed appropriately and rigorously? 

Reviewer #1: Yes

3. Have the authors made all data underlying the findings in their manuscript fully available?

Reviewer #1: Yes

4. Is the manuscript presented in an intelligible fashion and written in standard English?

Reviewer #1: No

5. Review Comments to the Author

Reviewer #1: Major

1. Overall, the manuscript needs to be written in a more concise manner. The abundance of run on sentences is distracting

2. What database are the sequences uploaded to for public access?

3. In the Faecal and caecal microbiome analysis section, what were the quality control parameters to screen the DNA reads before analysis? Also, what region of 16S rRNA was amplified? Line 161 mentions denoising and quality filtering chimera checking, add specifics to the methods.

4. What is the rationale for using both PCoA and PCA in results?

5. Discussion needs more clarity. Removing run-on sentences. For example, line 274-277 needs to be shortened sentence. The purpose is lost with the abundance of words.

6. Conclusion needs to be rewritten to be more concise. Ideally sentence is a run-on and it take away from the significance of the conclusions.

Minor

1. Revise of the title to be more concise

2. Define what “gut tract” represents early on in abstract

3. Line consider changing “from the same animal or from a different animal” to individual variation to make more concise

4. Line 61-64 confusing, consider revising sentence

5. Line 67 remove “anyhow”

6. Line 111 remove “from the”

7. Within the results lines 240, 242, 243 mention % of relative abundance, for consistency please add % to remainder of bacterial genera mention throughout that paragraph

8. Line 278 remove “in fact”

9. Line 284 remove "however”

10. Line 285 remove “Having ascertained that”

11. Consider adding the function of the microbiome in both cecum and colon to discussion to add to your relevance.

12. Purpose of paragraph 297-300 is unclear, consider removing or changing

13. Line 307 another is spelled wrong

14. Paragraph 370-374 also need clarification, unclear purpose

6. PLOS authors have the option to publish the peer review history of their article (what does this mean?). If published, this will include your full peer review and any attached files.

Reviewer #1: No

---

## [Author Response · Author response to Decision Letter 0]

2 Jan 2024

Reviewer #1: Major

1. Overall, the manuscript needs to be written in a more concise manner. The abundance of run on sentences is distracting

ANSWER: We have revised the whole text breaking convoluted sentences in shorter and clearer.

2. What database are the sequences uploaded to for public access?

ANSWER: The sequences have been deposited in the GenBank https://www.ncbi.nlm.nih.gov/ SRA (Sequence Reads Archive repository under the code PRJNA975099.

The information has been added to the manuscript.

3. In the Faecal and caecal microbiome analysis section, what were the quality control parameters to screen the DNA reads before analysis? Also, what region of 16S rRNA was amplified? Line 161 mentions denoising and quality filtering chimera checking, add specifics to the methods.

ANSWER: as regards the target regions of the 16S rRNA, a multi-amplicon sequencing strategy) targeting V2, V3, V4, V6-7, V8 and V9 hypervariable regions was adoptedusing the Ion 16STM Metagenomics multi-amplicon Kit (Thermo Fisher Scientific, Waltham, MA). 

The specification has been included in the text.

Further, the requested detail on quality checks has been added as follows:

The uBAM files sourced from the Ion GeneStudio platform were converted into FASTQ format using the samtools bamtofastq (v1.10). A 20-nucleotide trimming on both ends of the raw reads was performed to eliminate the sequencing primers using cutadapt (v3.5) A “Quantitative Insights Into Microbial Ecology 2” (QIIME2) (v2020.08) pipeline was subsequently used to analyse the trimmed raw reads. Within this process, imported reads were denoised and dereplicated using the “qiime dada2” plugin followed by OTU clustering with a 97% sequence similarity cutoffs using the Qiime vsearch plugin. The representative sequences from OTUs were then classified using SILVA SSU (version 138.1) as the reference database. 

4. What is the rationale for using both PCoA and PCA in results?

ANSWER: While in case the Euclidean measure were used as the basis of ordination for both methods, their outputs would be equivalent, when adopting Bray-Curtis distances, as in the present case for the PCoA, the two types of plots provide different complementary aspects. PCA emphasizes the linear correlation aspects while PCoA is better visualizing the distances that separate each sample from the others.

5. Discussion needs more clarity. Removing run-on sentences. For example, line 274-277 needs to be shortened sentence. The purpose is lost with the abundance of words.

ANSWER: We split those lines in two sentences with a period and rephrased the beginning of the first one..

6. Conclusion needs to be rewritten to be more concise. Ideally sentence is a run-on and it take away from the significance of the conclusions.

ANSWER: We re-wrote the conclusion in concise sentences as follows: 

“Three variables were studied in relation to the fattening quail gut microbiome. These included anatomical location, administered diet, and single individual identity. The hierarchical order by which they act in determining bacterial community structure was assessed. The ranking indicated the gut compartment as the ruling determinant, with the diet type as second force, and, in suborder, the individual animal relay.”

Minor

1. Revise of the title to be more concise

ANSWER: The title has been shortened by removing the words “the variables of”.

2. Define what “gut tract” represents early on in abstract

ANSWER: Correction made (comparing caecum to rectum)

3. Line consider changing “from the same animal or from a different animal” to individual variation to make more concise

ANSWER: We shortened the expression (from the same individual).

4. Line 61-64 confusing, consider revising sentence

ANSWER: The sentence has been divided in two separate ones (“Factors that have been demonstrated effective in affecting microbiome composition include environmental ones, as biogeographical context and population social structure. Others are related to the specific individual, including sex and reproductive condition.”)

5. Line 67 remove “anyhow”

ANSWER: Correction made.

6. Line 111 remove “from the”

ANSWER: Correction made.

7. Within the results lines 240, 242, 243 mention % of relative abundance, for consistency please add % to remainder of bacterial genera mention throughout that paragraph

ANSWER: Correction made.

8. Line 278 remove “in fact”

ANSWER: Correction made.

9. Line 284 remove "however”

ANSWER: Correction made.

10. Line 285 remove “Having ascertained that”

ANSWER: Correction made.

11. Consider adding the function of the microbiome in both cecum and colon to discussion to add to your relevance.

ANSWER: The suggested text has been added as follows: “To frame the focus of this comparison it is worth recalling that in birds, the role of caeca involves not only the reabsorption of water and salts but also fermentation of undigested carbohydrates and uric acid, thereby re-cycling nitrogen and generating ammonia and volatile fatty acids. Conversely, the rectum, from which the faeces are ejected is a terminal passage with no specific physio-metabolic functions.”

12. Purpose of paragraph 297-300 is unclear, consider removing or changing

ANSWER: It was as disclaimer to back up the choice of discussing results at species level rather than limiting to the genus rank. We modified the sentence to clarify it.

13. Line 307 another is spelled wrong

ANSWER: Correction made.

14. Paragraph 370-374 also need clarification, unclear purpose

ANSWER: We agree and the sentence was actually not necessary. We removed it.

Having addressed all the pending issues, and thanking the Reviewer for having contributed to a substantial improvement of this manuscript, we look forward to hearing from you.

---

## [Decision Letter · Decision Letter 1]

23 Jan 2024

Determining the hierarchical order by which intestinal tract, administered diet, and individual relay can shape the gut microbiome of fattening quails

PONE-D-23-33570R1

Dear Dr. Andrea Squartini,

We’re pleased to inform you that your manuscript has been judged scientifically suitable for publication and will be formally accepted for publication once it meets all outstanding technical requirements.

Kind regards,

Ewa Tomaszewska, DVM Ph.D

Academic Editor

PLOS ONE

Additional Editor Comments (optional):

Reviewers' comments:

Reviewer's Responses to Questions

**Comments to the Author**

1. If the authors have adequately addressed your comments raised in a previous round of review and you feel that this manuscript is now acceptable for publication, you may indicate that here to bypass the “Comments to the Author” section, enter your conflict of interest statement in the “Confidential to Editor” section, and submit your "Accept" recommendation.

Reviewer #1: All comments have been addressed

2. Is the manuscript technically sound, and do the data support the conclusions?

Reviewer #1: Yes

3. Has the statistical analysis been performed appropriately and rigorously? 

Reviewer #1: Yes

4. Have the authors made all data underlying the findings in their manuscript fully available?

Reviewer #1: Yes

5. Is the manuscript presented in an intelligible fashion and written in standard English?

Reviewer #1: Yes

6. Review Comments to the Author

Reviewer #1: Authors addressed all original comments and made appropriate edits. Manuscript is ready to proceed. Thank you!

7. PLOS authors have the option to publish the peer review history of their article (what does this mean?). If published, this will include your full peer review and any attached files.

Reviewer #1: No

---

## [Editor Report · Acceptance letter]

11 Mar 2024

PONE-D-23-33570R1 

PLOS ONE

Dear Dr. Squartini, 

I'm pleased to inform you that your manuscript has been deemed suitable for publication in PLOS ONE. Congratulations! Your manuscript is now being handed over to our production team.

Kind regards, 

on behalf of

Professor Ewa Tomaszewska 

Academic Editor

PLOS ONE